# In Vitro Anticancer Activity of Methanolic Extract of *Justicia adhatoda* Leaves with Special Emphasis on Human Breast Cancer Cell Line

**DOI:** 10.3390/molecules27238222

**Published:** 2022-11-25

**Authors:** Sonu Kumar, Rajveer Singh, Debrupa Dutta, Shivani Chandel, Arka Bhattacharya, Velayutham Ravichandiran, Soumi Sukla

**Affiliations:** 1Department of Pharmacology & Toxicology, National Institute of Pharmaceutical Education and Research, 168, Maniktala Main Road, Kolkata 700054, India; 2Department of Natural Products, National Institute of Pharmaceutical Education and Research, 168, Maniktala Main Road, Kolkata 700054, India

**Keywords:** *Justicia adhatoda*, natural product, leaf extract, anticancer, MCF-7, breast cancer, apoptosis, NF-κB

## Abstract

Natural products are being targeted as alternative anticancer agents due to their non-toxic and safe nature. The present study was conducted to explore the in vitro anticancer potential of *Justicia adhatoda (J. adhatoda)* leaf extract. The methanolic leaf extract was prepared, and the phytochemicals and antioxidant potential were determined by LCMS analysis and DPPH radical scavenging assay, respectively. A docking study performed with five major alkaloidal phytoconstituents showed that they had a good binding affinity towards the active site of NF-κB. Cell viability assay was carried out in five different cell lines, and the extract exhibited the highest cytotoxicity in MCF-7, a breast cancer cell line. Extract-treated cells showed a significant increase in nitric oxide and reactive oxygen species production. Cell cycle analysis showed an arrest in cell growth at the Sub-G0 phase. The extract successfully inhibited cell migration and colony formation and altered mitochondrial membrane potential. The activities of superoxide dismutase and glutathione were also found to decrease in a dose-dependent manner. The percentage of apoptotic cells was found to increase in a dose-dependent manner in MCF-7 cells. The expressions of caspase-3, Bax, and cleaved-PARP were increased in extract-treated cells. An increase in the expression of NF-κB was found in the cytoplasm in extract-treated cells. *J. adhatoda* leaf extract showed a potential anticancer effect in MCF-7 cells.

## 1. Introduction

Noncommunicable diseases (NCDs) such as cancer are widely acknowledged as a global health problem that lacks a universal solution [1]. Breast cancer (BC) is the primary cause of premature death in women among the various types of cancer. In the United States, breast cancer accounts for the second most cancer deaths in women after lung cancer. BC can be triggered by a range of risk factors such as race, ethnicity, family history of cancer, genetic features, and modifiable exposures such as alcohol consumption, physical inactivity, exogenous hormones, and certain female reproductive variables [2]. Breast cancer, when detected at early stages, is treatable, particularly when it is still limited to the breast areas or has only migrated to the axillary lymph nodes. Advancements in multimodal therapy have increased the chances of a cure in 70–80% of patients. The advanced (metastatic) stage of breast cancer, on the other hand, is considered to be untreatable with the present therapeutic options [3]. Despite significant advancements in cancer treatment tactics, adverse effects and resistance to currently available anticancer medications are among the leading causes of treatment failure in 80% of patients [4]. Chemotherapeutic agents are highly effective in treating breast cancer, but they must be used for an extended period to obtain optimum results. Currently, targeted breast cancer therapy includes tamoxifen, fulvestrant, herceptin, and aromatase inhibitors. However, chemotherapeutic drug resistance is the major hurdle to overcome [5]. In addition to curing cancer, it can induce adverse health conditions such as ovarian and endometrial hyperplasia and carcinomas, as well as multiple vein thrombosis and pulmonary embolism [6]. Chemotherapeutic agents attack malignant cells and eradicate them, but they can also harm healthy cells and tissues, resulting in a significant decrease in patient life expectancy. As a result, innovative, target-oriented, safe, and low-cost therapeutic strategies are needed. Medicinal plants are relatively safe and inexpensive. Therefore, they could be a good alternative for potential new anticancer compounds [7]. Natural products and phytomedicines are the biggest hope for curing various human degenerative diseases, including cancer. Plants have been investigated as a source of medicinally relevant bioactive components, and many of them, such as Taxol, Camptothecin, Vincristine, Vinblastine, Vinorelbine, Vindesine, Vinflunine, and others, appear to have a strong therapeutic value in the treatment of a variety of malignancies [8]. Several plant species, including *Taxus brevifolia Nutt*, *Curcuma longa*, *Podophyllum peltatum*, and *Catharanthus roseus*, have been demonstrated to have potent anticancer properties against various types of cancer including breast cancer [9,10].

*J. adhatoda* (Acanthaceae), commonly known as Vasaka or Malabar, is an important medicinal plant used in ayurvedic and different traditional medicine systems [11]. It is distributed throughout the South Asia and Indo-China region and is natively found in India, Afghanistan, Bangladesh, East Himalayas, Laos, Myanmar, Nepal, Pakistan, Sri Lanka, and Vietnam [12,13]. It was also introduced into China, Cuba, Ethiopia, Jamaica, Madagascar, and Sicilia [12,13]. Various components of the plant are used to cure asthma, joint pain, lumber pain, sprains, colds, coughs, eczema, malaria, rheumatism, swelling, and venereal infections [11,14]. Plant leaves have anti-inflammatory properties and are also used as cardiotonic, expectorant, and anti-asthma agents [15]. Various leaf mixtures are often used in Southeast Asia to treat bleeding, skin illnesses, wounds, headaches, snake bites, and leprosy [16,17]. *J. adhatoda* has been studied for its antioxidant, hepatoprotective, anti-inflammatory, antibacterial, antiprotozoal, hypolipidemic, antiasthmatic, and abortifacient properties [18]. Alkaloids, tannins, saponins, phenolics, and flavonoids are bioactive phytochemicals found in the leaves of *J. adhatoda*. Pyrroquinazoline alkaloids such as vasicine, vasicol, vasicinone, and peganine, as well as other minor elements, make up the majority of it [19,20].

In light of these observations, the current study has been taken up to evaluate the anticancer properties of *J. adhatoda* leaf extract with particular emphasis on MCF-7, a breast cancer cell line.

## 2. Results

### 2.1. Phytochemical Composition of Methanolic Extract of J. adhatoda

LC/MS analysis of the methanolic extract of *J. adhatoda* leaves showed 81 compounds belonging to different chemical classes (Appendix A and Appendix A).

### 2.2. DPPH Radical Scavenging Assay

The DPPH assay is used to predict antioxidant activities by mechanism in which antioxidants inhibit oxidation and therefore determine free radical scavenging activity. The method is based on the scavenging of DPPH radicals by antioxidants, which, upon a reduction reaction, decolorize the purple DPPH solution [21]. The extract showed mild antioxidant potential with a significant increase in free radical scavenging activity in a concentration-dependent manner, and 30% free radical inhibition was observed at 250 µg/mL of extract, the highest concentration studied (Figure 1).

### 2.3. Docking Study

The plant’s active alkaloidal constituents include vasicine, l-vasicinone, deoxyvasicine, maiontone, vasicinolone, and vasicinol, among others, which have been extensively explored for a variety of pharmacological uses. Previously these molecules were explored for various antibacterial [22] and anti-neurodegenerative disorders [23] in in silico models. Therefore, these phytoconstituents were chosen for the molecular docking investigation [24,25,26]. NF-κB is a well-known transcription factor for biological responses, immune regulation, and inflammation, but mounting evidence indicates it also participates in oncogenesis and promotes the growth and spread of breast cancer. NF-κB activity is seen in many lymphoid or myeloid tumors, including multiple myeloma, Hodgkin diseases, and some non-Hodgkin lymphomas [27]. This transcription factor can, in most cases, protect transformed cells from apoptosis and thus participate in the onset or progression of many human cancers. Additionally, many solid tumors, including breast malignancies, glioblastomas, and many others, have NF-κB overexpression that makes them resistant to chemotherapeutic drugs such as tamoxifen and paclitaxel [28,29]. Hence, NF-κB is thought to be a target for drugs that fight breast cancer, and the molecular docking study with five major alkaloidal phytoconstituents of the extract showed significant binding affinity towards the active site NF-κB and can act as NF-κB inhibitor (Figure 2). The binding energy of the individual compounds is given in Table 1.

### 2.4. Evaluation of the Cytotoxicity of J. adhatoda Leaf Extract on Different Cell Lines

The anticancer activity of *J. adhatoda* leaf extract was evaluated by MTT assay on different cell lines, namely HEK-293, RAW 264.7, SHSY-5Y, MCF-7, and A549 after 24 h of treatment with different concentrations of *J. adhatoda* leaf extract (50 µg/mL, 100 µg/mL, 150 µg/mL, 200 µg/mL and 250 µg/mL) (Appendix A). The extract has shown significant decline in % cell viability with an IC_50_ value of 257.44 µg/mL in RAW264.7, 364.57 µg/mL in SHSY-5Y, 161.57 µg/mL in MCF-7 and 170.12 µg/mL in A549 (Figure 3A). The IC_50_ value for HEK-293 was found to be much higher, i.e., 695.86 µg/mL, suggesting that the extract is less cytotoxic on non-cancerous cell lines. According to our findings, the extract has shown the most cytotoxic effect on the MCF-7 cell line (Figure 3A). Standard drug paclitaxel showed concentration-dependent cytotoxicity in MCF-7 cells post 24 h treatment (Figure 3B).

### 2.5. J. adhatoda Leaf Extract Induces NO Production in MCF-7 Cell Line

Nitric oxide (NO) is an important cell signaling molecule. NO has a dual role in cancer biology; at concentrations of more than 500 nM, nitrogen species were found to possess a cytotoxic effect [30]. NO species induces cell death by causing DNA damage. The result showed a significant increase in NO production in MCF-7 cells treated with *J. adhatoda* leaf extract and standard drug paclitaxel as compared to control cells (Figure 4).

### 2.6. J. adhatoda Leaf Extract Induces ROS Production in MCF-7 Cell Line

In cellular physiopathology, ROS play a crucial role in cell proliferation, signaling pathways, and oxidative defense systems that kill microorganisms and cancerous cells. For this reason, the measurement of intracellular ROS production could represent a very useful parameter to quantify oxidative stress and its significance as an anticancer agent. Intracellular ROS quantification using fluorescent DCFDA dye showed that MCF-7 cells treated with *J. adhatoda* leaf extract resulted in more ROS production in a concentration-dependent manner than control cells with extract concentration of 150 µg/mL and 0.1 µM paclitaxel exhibiting similar ROS production (Figure 5A).

### 2.7. Effect of J. adhatoda Leaf Extract on Mitochondrial Membrane Potential (ΔΨM)

Mitochondria are involved in cell apoptosis, and the mitochondrial membrane potential (ΔΨM) is an important parameter to determine the mitochondrial function that can be utilized as an indicator of cell health. The integrity of the mitochondrial membrane was investigated in MCF-7 cells using JC-1 stain after treatment with *J. adhatoda* leaf extract. JC-1 accumulates and oligomerizes in the mitochondrial matrix of healthy cells, leading to the production of red fluorescence. Whereas in unhealthy cells having reduced mitochondrial membrane potential, JC-1 appears as a monomer and emits green fluorescence. The red/green fluorescence ratio is a measurement of mitochondrial membrane potential. The results obtained from confocal microscopy imaging showed a decreased red/green fluorescence intensity in *J. adhatoda* treated MCF-7 cells compared to control cells, indicating *J. adhatoda* induces mitochondrial membrane depolarization (Figure 5B,C).

### 2.8. Effect of J. adhatoda Leaf Extract on Glutathione (GSH) and Glutathione Disulfide (GSSG) Level

Malignant cells have a high metabolic activity, which results in the formation of a lot of free radicals. Under oxidative stress, GSH reduces to form GSSG. The ratio of GSH/GSSG is inversely proportional to the level of oxidative stress. GSH in cancerous cells elevates to maintain the balance of oxidative stress. The result obtained by treating MCF-7 cells with *J. adhatoda* leaf extract depicts that the extract reduces intracellular GSH levels (Figure 5D), which further results in the generation of ROS and induction of apoptosis in MCF-7 cells.

### 2.9. Effect of J. adhatoda Leaf Extract on Superoxide Dismutase Activity

Superoxide dismutase plays a crucial role in scavenging free radicals within cells, but oxidative radicals also significantly eliminate pathogens and cancerous cells. A plot of % SOD activity with an increase in *J. adhatoda* leaf extract concentrations at different time points (0, 10, 20, 30, 40, 50, and 60 min) revealed that *J. adhatoda* leaf extracts lowered % SOD activity in a concentration-dependent manner (Figure 5E).

### 2.10. J. adhatoda Leaf Extract Induces Apoptosis

Apoptosis is a standard physiologic procedure of controlled cell death to maintain stable homeostasis. Annexin V-FITC and propidium iodide (PI) double staining method showed that cells treated with *J. adhatoda* leaf extract (100 μg/mL and 150 μg/mL) induced early and late apoptosis compared to control cells (Figure 6A). These findings indicate that *J. adhatoda* induces apoptosis in a concentration-dependent manner in MCF-7 cells.

### 2.11. J. adhatoda Leaf Extract Induces Sub-G0 and S-Phase Cell Cycle Arrest

Cell cycle arrest is one of the most important methods by which anticancer drugs inhibit tumor progression. Generally, the G0/G phase is characterized by 2n (diploid), the G2/M phase by 4n (tetraploid), and the S phase between 2n and 4n. The impact of *J. adhatoda* leaf extract treatment on cell cycle phase propagation of the MCF-7 cells was evaluated. The results showed normal distribution of the G1, S, and G2/M phase in control cells. In contrast, cells treated with *J. adhatoda* leaf extract showed significant cell arrest at the Sub-G0 phase of the cell cycle (Figure 6B).

### 2.12. Effect of J. adhatoda Leaf Extract on Apoptosis-Related Protein Expression

Caspase-3, Bax, and cleaved-PARP are the major apoptosis-related protein molecules; hence measuring their expression in cells provides crucial data on apoptosis. The expression of these proteins in MCF-7 cells treated with *J. adhatoda* leaf extract was determined by Western blot analysis (Figure 6C). The results demonstrate that the expression of apoptotic proteins (Bax, caspase-3, and cleaved-PARP) was elevated compared to the control (Figure 6D).

### 2.13. Evaluation of J. adhatoda Leaf Extract on Colony Formation of MCF-7 Cells

Cancer cells grow in colonies in conjunction with adjacent cells; losing contact with adjoining cells leads to their death. The results obtained from the clonogenic assay (Figure 7A) showed a decrease in colony formation in *J. adhatoda* leaf extract-treated MCF-7 cells compared to that of control cells. The colonial/population density of each plate was measured and plotted (Figure 7B). The results showed that the colony growth was decreased by approximately 50% and 80% in cells treated with 100 μg/mL and 150 μg/mL of *J. adhatoda* leaf extract, respectively, compared to control cells. The result supports the anticancer role of *J. adhatoda* leaf extract.

### 2.14. J. adhatoda Leaf Extract Showed Inhibitory Activity in Cell Migration

MCF-7 control cells exhibited wound-healing properties and cell migration, and the gap was almost closed. On the contrary, significant inhibition of wound healing or cell migration was observed in the MCF-7 cells treated with *J. adhatoda* leaf extract (at concentrations of 100 μg/mL and 150 μg/mL) (Figure 7C). Percentage wound closure significantly declined in a concentration-dependent manner (Figure 7D). These results suggest that *J. adhatoda* leaf extract possesses anticancer properties.

### 2.15. Evaluation of J. adhatoda Leaf Extract on NF-κB Translocation

NF-κB is a well-known transcription factor in biological responses, immunological control, and inflammation, but accumulating evidence suggests it also plays a role in oncogenesis. Immunostaining of *J. adhatoda* leaf extract-treated MCF-7 cells using an anti-NF-κB antibody indicate that NF-κB translocated from cytoplasm to nucleus in control cells, but after treatment with *J.adhatoda* leaf extract at concentrations 100 μg/mL and 150 μg/mL, NF-κB expression was increased in the cytoplasm (Figure 8).

## 3. Discussion

Breast cancer is one of the most common cancers in women and is the leading cause of female death globally. BC is characterized by tumor heterogeneity, which leads to drug resistance to chemotherapy [3]. In developed countries, advances in diagnostic techniques and awareness of mammographic screening have significantly lowered the number of BC cases, but this is still an issue in developing countries [31]. It is curable if detected early or at the non-metastatic stage; however, it is incurable if detected later or after the spread of the metastasis using the currently available chemotherapeutic agents [4]. Treatment of BC is becoming increasingly difficult due to major side effects associated with existing therapies such as chemotherapy and radiotherapy. Currently, multidrug resistance to chemotherapy has made it difficult to treat BC patients, resulting in an increase in the number of cases as well as a decrease in patient survival rates [32]. To tackle this life-threatening illness, numerous research projects are currently committed to discovering alternative BC therapy systems that can be employed as therapies or adjuvant treatments alongside chemotherapeutic agents [33]. Natural compounds have attracted the scientific community’s attention in the hunt for more effective and less toxic medications for the treatment of cancer, particularly breast cancer, and have opened up a new area in chemotherapeutic research [34]. Curcumin derived from turmeric, resveratrol derived from grapes, and quercetin derived from citrus fruits have been demonstrated to have anticancer effects in pancreatic and colorectal cancer [35]. Many natural compounds extracted from herbs, such as paclitaxel [36], vincristine [37], cantharidin sodium injection [38], and Magnolol [39], have been demonstrated in clinical studies to have strong anti-breast cancer activity [40]. In this study, we explored the antiproliferative activity and mechanism of action of *J. adhatoda* leaf extract in the MCF-7 breast cancer cell line. *J. adhatoda* leaf extract did not show any significant antioxidant activity and was effective in killing tumor cells (RAW 264.7, SHSY-5Y, A549, and MCF-7) while showing a minimal effect on non-cancerous cells (HEK-293). Among all the tested cell lines, *J. adhatoda* has shown significant toxicity against the breast cancer cell line (MCF-7). Therefore, the breast cancer cell line MCF-7 was chosen to test further anticancer properties of the extract. The clonogenic assay showed that *J. adhatoda* leaf extract promoted breast cancer cell death and inhibited colony formation in an anchorage-independent manner. Cell migration is a key phase in the development and metastasis of cancer, and cancer metastasis is linked to the stimulation of cancer cell migration and invasion of the nearby tissues making it important to study and understand in order to combat the disease [41,42]. Furthermore, as demonstrated by a wound-healing assay, the extract inhibited cell migration and proliferation. The production of ROS and reactive nitrogen species (RNS) is required for cancer cells to grow, proliferate, and spread. Due to the higher metabolic rate, malignant cells produce more ROS and RNS. Depending on their local concentration, these species play pleiotropic functions in cancer and can cause tumor development or tumor suppression [43]. Simultaneously, excess ROS and RNS production over the required amount may cause oxidative stress in cancer cells, leading to death and a reduction in chemotherapy resistance [44,45]. In order to check the activity of the extract against ROS and RNS production, ROS assay by DCFDA method and NO assay by Griess reagent were performed. The results obtained from these assays suggest that *J. adhatoda* leaf extract produced both ROS and RNS in considerable amounts and in a concentration-dependent manner in the MCF-7 cell line and that paclitaxel treatment also did the same [46]. The extract’s anticancer properties are probably related to the presence of phytochemicals disrupting the MCF-7 cells’ redox balance, which is necessary for survival. After exposure to *J. adhatoda* leaf extract, higher percentages of apoptosis and cell cycle arrest at Sub-G0 and S-phase have been detected in a concentration-dependent manner. Apoptosis was further confirmed by Western blot analysis of apoptotic markers (Caspase-3, Bax, and cleaved-PARP). NF-κB is a vital transcription factor that links inflammation to cancer, and previous data suggest that NF-κB has a role in breast cancer tumorigenesis and chemotherapeutic resistance [47]. The expression of NF-κB p65 declined in the nucleus of MCF-7 cells after treatment with typical chemotherapeutic medicines (Tamoxifen) in breast cancer cells [48]. The current study also showed that the extract dramatically reduced NF-κB expression in the nucleus of MCF-7 cells in a concentration-dependent manner compared to the control group. Docking studies confirmed that the major alkaloidal phytoconstituents of the extracts, such as vasicine, vasicinol, vasicinone, deoxyvasicinone, and vasicinolone, have a significant binding affinity towards the active site of NF-κB and can act as NF-κB inhibitors. From the above findings, it can be stated that *J. adhatoda* leaf extract has anti-breast cancer potential because it can induce apoptosis by activating apoptotic proteins (caspase-3, Bax, and cleaved-PARP), prevent cells from entering in Sub-G0 phase of the cell cycle and inactivate the NF-κB pathway. Overall, our findings suggest that *J. adhatoda* would be an effective treatment for chemotherapeutic-resistant breast cancer. Further pre-clinical studies on *J. adhatoda* leaf extract in animal models of breast cancer are warranted to check its chemotherapeutic potential.

## 4. Materials and Methods

### 4.1. Preparation of Mathanolic Extract of J. Adhatoda Leaves

*J. adhatoda* plants were collected from the West Bengal State Medicinal Plants Board, Kolkata, washed thoroughly, and shade dried at room temperature. The dried leaves were ground with a mechanical grinder to make a coarse powder and were sieved (mesh size #40) to separate large particles [12]. Further, 100 g of dried powder was macerated in 500 mL of methanol for 15 days with continuous shaking, followed by filtration with Whattman No. 1 filter paper. The filtrate was concentrated using a rotary evaporator (Heidolph Instruments GmbH & CO. KG, Schwabach, Germany). The concentrated extract was vacuum dried and stored at 4 °C [49].

### 4.2. Analysis of Plant Extract by LC-MS

Chromatographic separation (Agilent Technologies 1290 Infinity ıı, Agilent Technologies, Waldbronn, Germany) was achieved on an ACQUITY UPLC^®^HSS C18 column (2.1 × 100 mm, 1.8 μm). The mobile phase consisted of acetonitrile as solvent A, methanol as solvent B, and 0.5% acetic acid in water as solvent C (Appendix A). The flow rate was 0.6 mL/min, and the column temperature was set at 20 °C. The diode-array detection was set to monitor at 270 nm. The mass spectrometry analysis was performed using a 6545 Q-TOF LC/MS coupled mass spectrometer, with the following conditions: source temperature: 350 °C; ion spray voltage −4.5 kV; Gas 1, Gas 2, curtain gas, and collision gas (nitrogen) were separately set at 50, 50, 45, and 12 psi; sheath gas flow rate, 40 arbitrary units; auxiliary gas flow rate, 5 arbitrary units; electrospray voltage, 3.5 kV; capillary voltage, −32 V; capillary temperature, 270 °C [50].

### 4.3. DPPH Free Radical Scavenging Activity

To assess the antioxidant potential of *J. adhatoda* leaf extract, DPPH radical scavenging assay was conducted according to Naqvi et al. with few modifications [51,52]. Briefly, 24 mg of 2,2-diphenyl-1-picrylhydrazyl (DPPH) reagent was dissolved in 100 mL methanol, and the absorbance was checked to be 1.08–1.1 at 517 nm. Then, 100 µL of extract in concentrations of 50 µg/mL, 100 µg/mL, 150 µg/mL, 200 µg/mL, and 250 µg/mL was added to 100 µL DPPH solution and incubated for 20 min at room temperature in dark. The absorbance was checked at 517 nm. % inhibition of free radicals was calculated using the following formula:%inhibition = [(absorbance_control_ − absorbance_sample_)/absorbance_control_] × 100

### 4.4. Molecular Docking of the J. adhatoda Leaf Extract towards NF-κB

The list of phytochemicals present in *J. adhatoda* was prepared from the LC-QTOF-MS data. A total of 81 compounds were identified, and from them, 5 most active compounds were selected for docking study. The three-dimensional (3D) conformers of these compounds were downloaded from the PubChem database in SDF format. Similarly, the 3D structure of protein NF-κB (PDB ID: 1LE5) was acquired from the protein data bank (https://www.rcsb.org/, (accessed on 15 June 2022)). The 3D structures of the ligands were optimized with LigPrep (Schrödinger Release 2022-2: LigPrep, Schrödinger, LLC, New York, NY, USA, 2021), and receptor structure was optimized and prepared for docking using Protein Preparation Wizard. The active site of the proteins was evaluated by SiteMap, and the grid was generated through the Receptor Grid generator. Further, the optimized ligand and the receptor were docked using Ligand Docking in Glide-V9.2. The probable poses were observed through Pose Viewer. The Docking analysis was performed in Schrödinger Release 2022-2: Maestro, Schrödinger, LLC, New York, NY, USA, 2021.

### 4.5. Cell Culture

RAW 264.7 cells (murine macrophage cell line), HEK-293 cells (Human embryonic kidney cell line), SHSY-5Y cells (neuroblastoma cell line), MCF-7 cells (Michigan Cancer Foundation-7 cell line), and A549 cells (adenocarcinomic human alveolar basal epithelial cell line) were obtained from NCCS, Pune, India and maintained in Dulbecco’s modified Eagle’s medium (DMEM) supplemented with 10% (*v*/*v*) fetal bovine serum (FBS) and penicillin-streptomycin (1% *v*/*v*) (Gibco, Grand Island, NY, USA). The cells were grown in a humidified incubator at 37 °C with 5% carbon dioxide [53].

### 4.6. Measurement of Cell Viability

HEK-293 cells, RAW 264.7 cells, SHSY-5Y cells, MCF-7 cells, and A549 cells were seeded in a 96-well plate (1 × 10^4^ cells/well) containing DMEM media supplemented with 10% FBS and incubated for 24 h. Cells were treated with different concentrations of *J. adhatoda* leaf extract (50 µg/mL, 100 µg/mL, 150 µg/mL, 200 µg/mL and 250 µg/mL) and paclitaxel (Sigma) with 0.01 µM, 0.1 µM, and 1 µM concentrations are used as positive control and incubated at 37 °C with 5% CO_2_ for 24 h. Further, 10 µL of MTT (3-(4, 5-dimethyl thiazolyl-2)-2, 5-diphenyltetrazolium bromide) reagent (5 mg/mL in 1X phosphate buffer saline) was added to each well, and cells were incubated for 4 h at 37 °C in 5% CO_2_. Subsequently, cells were washed with 1X PBS, and the insoluble formazan crystals were solubilized with 200 μL of dimethyl sulphoxide (DMSO). Absorbance was measured by using a microplate reader (Synergy H1 microplate reader, Biotek, Santa Clara, CA, USA) at 570 nm [54].
% viable cells = (OD of sample − OD of blank)/(OD of control − OD of blank) × 100

### 4.7. Nitric Oxide (NO) Production Assay

For quantification of NO production, MCF-7 cells were seeded in 6 well plates (3 × 10^5^ cells/well) and incubated for 12 h at 37 °C in a 5% CO_2_ incubator [55]. The media from each well was removed, and the cells were treated with 2 mL *J. adhatoda* leaf extract in serum-free media at concentrations of 100 µg/mL and 150 µg/mL, and 0.1 µM of paclitaxel treatment was used as standard positive control. After 1 h of treatment, the presence of nitrite was determined by taking the cell supernatant using a commercial NO detection kit (Sigma-Aldrich, St. Louis, MO, USA). Briefly, 50 μL of cell supernatant with an equal volume of Griess reagent was incubated in a 96-well plate at room temperature for 30 min [50]. Absorbance was measured by a microplate reader (Synergy H1 microplate reader, Biotek, Santa Clara, CA, USA) at 540 nm. The amount of nitrite in the media was calculated from the sodium nitrite (NaNO_2_) standard curve [56,57].

### 4.8. Measurement of Intracellular ROS Generation by DCFDA

For the quantification of intracellular reactive oxygen species (ROS) production, MCF-7 cells were seeded in 12-well plates (1 × 10^5^ cells/well) and incubated for 24 h [58]. The cells were then treated with 100 µg/mL and 150 µg/mL concentrations of *J. adhatoda* leaf extract, and 0.1 µM of paclitaxel treatment was used as standard positive control for 24 h. After 24 h of treatment, cells were incubated with 10 μM DCFDA for 30 min at 37 °C in the dark. Further, cells were detached using 0.25% trypsin-ethylenediaminetetraacetic acid (EDTA), centrifuged at 500× *g* for 5 min, and resuspended in 1X PBS. The flow cytometric analysis was performed using BD LSRFortessa^TM^ Cell Analyzer (Becton, Dickinson and Company, San Jose, CA, USA) [59], and data were analyzed using FACS analysis software (FlowJo^TM^ v 10.8 BD bioscience, San Jose, CA, USA).

### 4.9. Measurement of Mitochondrial Membrane Potential (ΔΨM) by JC-1

For mitochondrial membrane potential analysis, MCF-7 cells were plated in a confocal plate (1 × 10^4^ cells/well) and incubated overnight. The 80–90% confluent cells were treated with 100 µg/mL and 150 µg/mL concentration of *J. adhatoda* leaf extract for 24 h. After 24 h of treatment, the cells were washed with 1X PBS, and cells were re-incubated in fresh culture media with 2 µL of JC-1 dye (5 mg/mL) for 30 min at 37 °C. Media was removed, and the cells were washed 3 times with warm 1X PBS. Then, 1 mL of 1X PBS was added to each plate and covered with a sheet of aluminum foil. The fluorescence of the test culture was captured by using a confocal microscope (STELLARIS5, Leica Biosystems, Wetzlar, Germany) with 100X magnification [60].

### 4.10. Quantitative Determination of Glutathione (GSH) and Glutathione Disulfide (GSSG)

For quantification of the reduced sulfhydryl form (GSH) and the oxidized glutathione disulfide form (GSSG), MCF-7 cells were seeded in a 6-well plate (3 × 10^5^ cells/well) and incubated for 24 h. Next, cells were treated with 100 µg/mL and 150 µg/mL of *J. adhatoda* leaf extract for 24 h. After 24 h of treatment, cells were washed twice with 1X PBS and trypsinized using trypsin–EDTA for 5 min at room temperature (22–25 °C) [61,62]. Quantification of GSH and GSSG from cell lysate was performed using DetectX Glutathione (GSH) Fluorescent Detection kit (ArborAssays^TM^, Ann Arbor, MI, USA) following the manufacturer’s instructions, and fluorescence was measured by a microplate reader (Synergy H1 microplate reader, Biotek, Santa Clara, CA, USA) at 510 nm emission and 370–410 nm excitation wavelength, respectively.

### 4.11. Determination of the Superoxide Dismutase Activity

Superoxide dismutase (SOD) activity of *J. adhatoda* leaf extract (50 µg/mL, 100 µg/mL, 150 µg/mL, 200 µg/mL, and 250 µg/mL) was determined by SOD Determination Kit (Sigma-Aldrich, Catalogue no. 19160) following manufacturer’s instructions. The absorbance was measured using a microplate reader (Synergy H1 microplate reader, Biotek) at 450 nm to assess the SOD mimetic activity of *J. adhatoda* leaf extract in an aqueous solution at 25 °C. A plot of percentage SOD inhibition vs. *J. adhatoda* leaf extract concentration (µg/mL) at different time points was used to establish the IC50 value for WST (water-soluble tetrazolium salt) inhibition [63].

### 4.12. Quantification of Apoptotic Cells by Annexin V-PI

Apoptotic cell quantification was performed by annexin V-FITC and propidium iodide (PI) double staining method. MCF-7 cells were seeded into 12-well plates (1 × 10^5^ cells/well) and incubated for 24 h. Cells were treated with 100 µg/mL and 150 µg/mL of *J. adhatoda* leaf extract for 24 h. Next, the cells were washed twice with ice-cold 1X PBS and centrifuged at 500× *g* for 5 min at 4 °C. The cells were subjected to annexin V and PI staining, and the samples were run in a flow cytometer using BD LSRFortessa^TM^ Cell Analyzer. Apoptotic cells were quantified by using FACS analysis software (FlowJo^TM^ v 10.8 BD bioscience, San Jose, CA, USA).

### 4.13. Cell Cycle Analysis

Cell cycle analysis is essential in identifying the precise mechanism of cell arrest in a specific phase of the cell cycle [64]. For cell cycle analysis, MCF-7 cells were seeded in 12-well plates (1 × 10^5^ cells/well) and incubated at 37 °C with 5% CO_2_ for 24 h. The cells were treated with 100 µg/mL and 150 µg/mL concentrations of *J. adhatoda* leaf extract for 24 h. After 24 h, cells were centrifuged at 500× *g* for 5 min. The supernatant was discarded, and the pellet was redissolved in 70% ethanol. The cells were incubated at 4 °C overnight [65]. After incubation, the cells were centrifuged, and the cell pellet was washed twice with 1X PBS. Next, cells were resuspended in 100 μL of 1X PBS. RNAse was added, and the cells were incubated at 37 °C for 4 h. The centrifugation and washing steps were repeated. The cells were resuspended in 100 μL of 1X PBS, and 5 μL (5 mg/mL) of propidium iodide (PI) was added, and samples were incubated for 30 min at room temperature. The samples were run in a flow cytometer by using BD LSRFortessa^TM^ Cell Analyzer, and data were analyzed using FACS analysis software (FlowJo^TM^ v 10.8 BD bioscience, San Jose, CA, USA) [66].

### 4.14. Western Blot Analysis

MCF-7 cells were cultured for 24 h and treated with 100 µg/mL and 150 µg/mL concentrations of *J. adhatoda* leaf extract. Total protein from each sample was extracted using a radioimmunoprecipitation assay (RIPA Lysis and Extraction Buffer #89900) lysis solution, including phosphatase inhibitors. Using a Pierce™ BCA Protein Assay kit (Pierce™ BCA Protein Assay Kit #23225), the protein concentration was determined [67]. For each sample, an equivalent amount of protein was mixed with loading buffer to form an equal volume, and then the sample was denatured at 95 °C for 15 min. Further, the protein was separated on a 10% SDS-PAGE gel and transferred to the PVDF membrane [68]. The membranes were blocked for 2 h with blocking buffer (5% skim milk in 1X PBS buffer with 0.1% Tween 20), followed by incubation at 4 °C overnight with primary antibodies against β-actin (1:1000, Rabbit, Cell signaling #5174S), Bax (1:1000, Rabbit pAb, ABclonal #A19684), Caspase-3 (1:1000, Rabbit pAb, ABclonal #A2156) and cleaved-PARP (poly ADP-ribose polymerase, 1:1000, Rabbit, Cell signaling #9542). Next, the membranes were washed with 1X PBST buffer twice at 5 min intervals. Further, the membranes were incubated with horseradish peroxidase (HRP)-conjugated secondary antibody (Goat Anti-Rabbit, 1:4000, ABclonal #AS014) for 1 hr. The protein expression levels were determined using enhanced chemiluminescence (ECL) with ChemiDoc TM XRS+ (Biorad).

### 4.15. Clonogenic Assay

For the clonogenic assay, MCF-7 cells were seeded in 12-well plates. On reaching 80–90% confluency, cells were treated with 100 µg/mL and 150 µg/mL of *J. adhatoda* leaf extract for 24 h. After 24 h, the cells were trypsinized and seeded at a density of 5 × 10^3^ cells per well in a 12-well plate, followed by incubation for 14 days. Further, the cells were rinsed with 1X PBS and fixed in a 3.7% formaldehyde solution for 30 min. The fixed cells were then stained with bromophenol and incubated for another 30 min. The plates were washed with 1X PBS and allowed to dry at room temperature. Images were captured, and colonies were counted manually [69].

### 4.16. Wound Healing/Scratch Assay/Cell Migration Assay

For wound healing (scratch) assay, MCF-7 cells were seeded in a 12-well plate and were allowed to reach confluency up to 80–90%. A uniform scratch was made on the monolayer in each well using a 2.5 μL sterile pipette tip. The cells were rinsed with sterile 1X PBS to remove debris before being treated with 100 µg/mL and 150 µg/mL *J. adhatoda* leaf extract for 24 h and 48 h. The images of the cells were acquired with the help of an inverted microscope (Nikon-SC600) with 10× magnification to record the data, and the % wound closure was measured using ImageJ (Software 1.53 s, Wayne Rasband, National Institutes of Health, Bethesda, MD, USA). The percentage of wound closure area was determined as
% wound closure = [(Area of wound at 0 h − Area of wound at 24 h and 48 h)/Area of wound at 0 h] × 100

The experiment was repeated in triplicate.

### 4.17. Translocation of NF-κB

For nuclear factor κB (NF-κB) translocation analysis, cells were plated in a confocal plate (1 × 10^4^ cells/well). After 24 h, the cells were treated with 100 µg/mL and 150 µg/mL of *J. adhatoda* leaf extract for 24 h. Next, the cells were washed with 1X PBS and fixed with 4% formaldehyde in 1X PBS for 15 min. The fixed cells were permeabilized with 0.5% Triton-X100 in 1X PBS for 15 min, followed by blocking with 3% BSA for 1 h. Further, the cells were incubated with anti-NF-κB p65 (Cell signaling #3033) primary monoclonal antibody overnight at 4 °C. The cells were then washed with 1X PBS and further incubated with a secondary antibody conjugated with Alexa Fluor 488 (1:500, Anti-Rabbit; Cell signaling #4412S) for another 1 h. Samples were washed with 1X PBS. Cells were stained with DAPI for 1 h at 37 °C, and images were acquired using a confocal microscope (STELLARIS5, Leica Biosystems, Wetzlar, Germany) with 100× magnification [70,71].

## 5. Conclusions

The current study was undertaken to check the potential of *J. adhatoda* leaf extract as an anticancer agent. In silico studies with the five major alkaloid phytoconstituents of the extract indicated significant binding affinity towards the active site of NF-κB and potential as an NF-κB inhibitor. The extract increased NO and ROS production, whereas superoxide dismutase and glutathione activities were decreased when MCF-7, a breast cancer cell line, was treated with it. The extract inhibited cell migration and colony formation and altered mitochondrial membrane potential. The extract showed anti-breast cancer potential, as it induces apoptosis by activating apoptotic proteins like caspase-3, Bax, and cleaved-PARP. The extract also prevented cells from entering in Sub-G0 phase of the cell cycle. All of these in vitro studies clearly indicated the anti-breast cancer potential of the extract. When compared with Paclitaxel, one of the currently available drugs to treat breast cancer, the extract showed similar results. Natural products, when used for treatments, show minimal adverse effects. Currently available breast cancer chemotherapeutics sometimes fail to provide desired results and show different side effects as well as resistance to chemotherapy. To overcome these problems, anticancer agents from natural sources are being investigated worldwide. It can be concluded from the findings of the current study that *J. adhatoda* leaf extract has a high potential as an alternative therapy for breast cancer. Further pre-clinical studies need to be conducted to develop it as an anti-breast cancer agent.

## Figures and Tables

**Figure 1 molecules-27-08222-f001:**
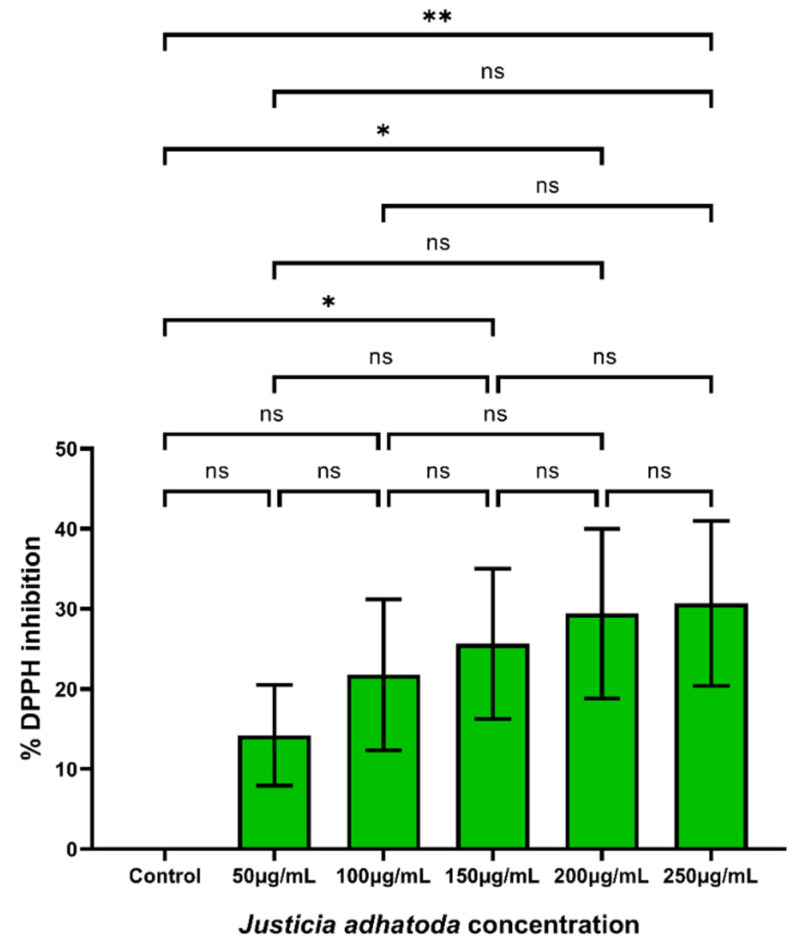
DPPH radical scavenging activity of *J. adhatoda* leaf extract. Different concentrations of *J. adhatoda* extract showed concentration-dependent increase in free radical scavenging activity by DPPH radical scavenging assay. One-way ANOVA test was used to perform the statistical analysis. Data represented as mean ± SD (*n* = 3), ** *p* < 0.01 vs. control; * *p* ≤ 0.05 and ns (non-significant) *p* > 0.05.

**Figure 2 molecules-27-08222-f002:**
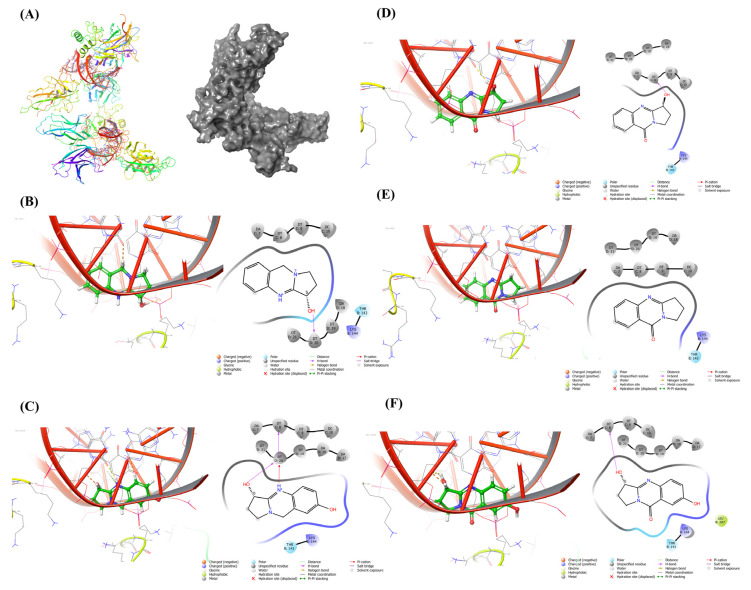
Docking study with NF-κB. Docking study was conducted to assess the binding affinity of major phytoconstituents of *J. adhatoda* with the active site of NF-κB. The (**A**) 3D structure of NF-κB protein (left) spiral structure (right) space-filling model. The poses are shown for the structures of (**B**) Vasicine (left) 3D interaction (right) 2D interaction, (**C**) Vasicinol (left) 3D interaction (right) 2D interaction, (**D**) Vasicinone (left) 3D interaction (right) 2D interaction, (**E**) Deoxyvasicinone (left) 3D interaction (right) 2D interaction, (**F**) Vasicinolone (left) 3D interaction (right) 2D interaction.

**Figure 3 molecules-27-08222-f003:**
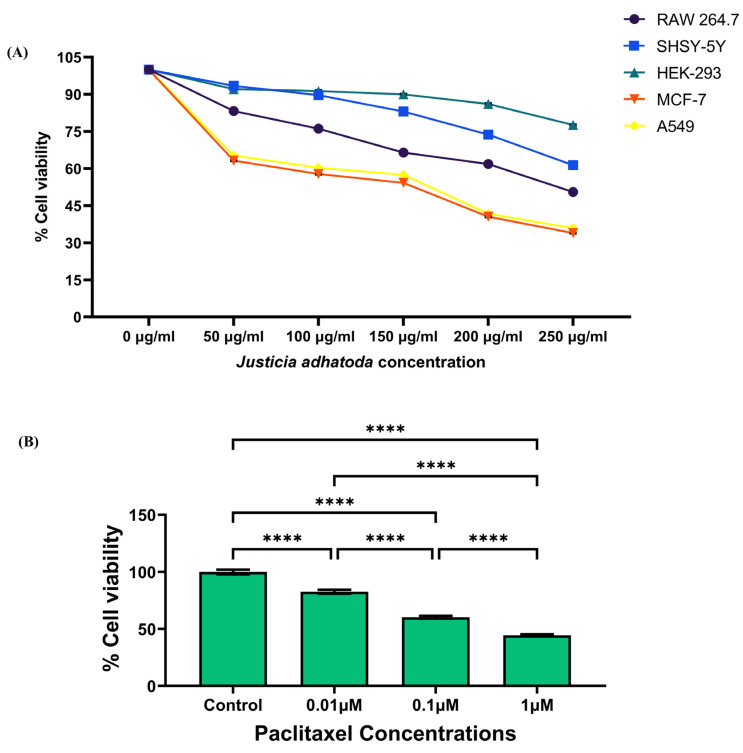
Determination of cell viability by MTT assay. Cell viability was assessed using the MTT assay for HEK-293 cells, RAW 264.7 cells, SHSY-5Y cells, MCF-7, and A549 cells treated for 24 h with various concentrations of *J. adhatoda* leaf extract: 50, 100, 150, 200, and 250 μg/mL (**A**) and for different concentrations of paclitaxel in MCF-7 cells post 24 h treatment as standard drug control (**B**). One-way ANOVA test was used to perform the statistical analysis, and the IC_50_ values of the extract on the cell lines were determined. Data represented as mean ± SD (*n* = 5), **** *p* < 0.0001 vs. control.

**Figure 4 molecules-27-08222-f004:**
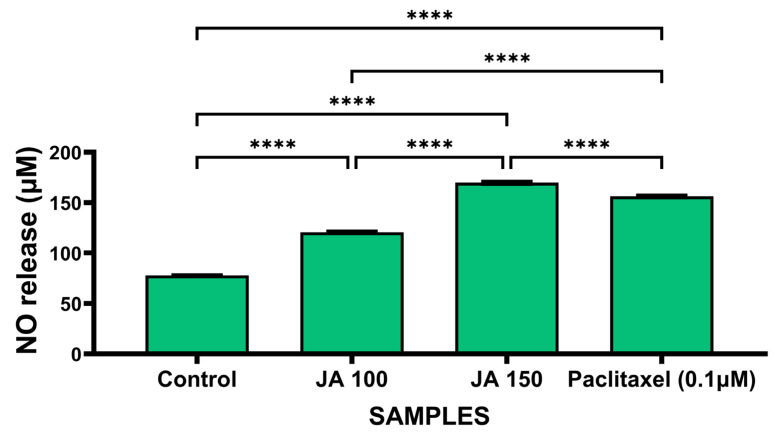
Quantification of NO production in MCF-7 cells by Griess reagent. μM of NO production from cell supernatant of MCF-7 cell treated with 100 μg/mL and 150 μg/mL *J. adhatoda* leaf extract and 0.1 µM paclitaxel as standard drug. The experiment was performed in triplicate and the result presented as mean ± SD (*n* = 3). **** *p* < 0.0001 vs. control.

**Figure 5 molecules-27-08222-f005:**
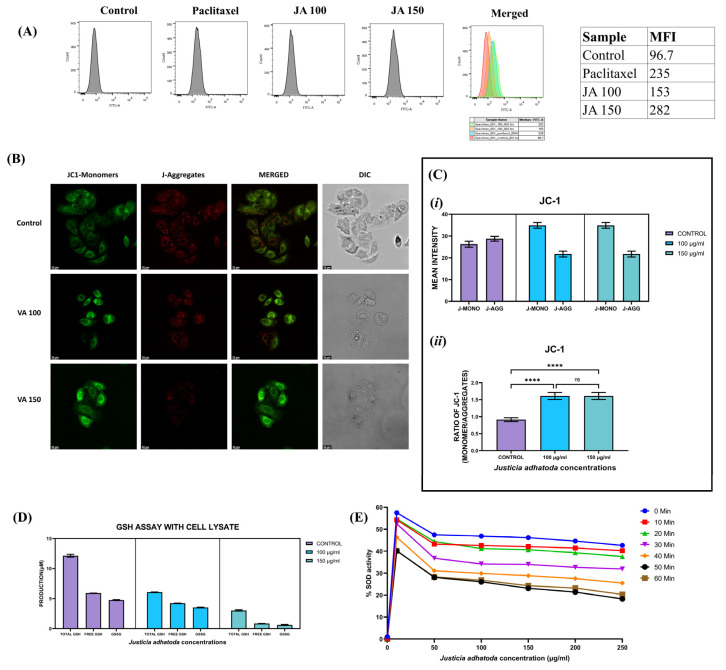
*J. adhatoda* leaf extract induced ROS production and mitochondrial membrane depolarization in MCF-7 cells. (**A**) MCF-7 cells showed elevated ROS production in a concentration-dependent manner as compared to control cells 24 h post-treatment with 100 μg/mL and 150 μg/mL of *J. adhatoda* leaf extract similar to 0.1 µM paclitaxel treated MCF-7 cells after staining with DCFDA. (**B**) Mitochondrial membrane potential was measured by confocal microscopy imaging (100x magnification) in MCF-7 cells stained with JC-1. Graphical data representing the (**C**) (*i*) mean fluorescence intensity of JC-1 monomer and JC-1 aggregates of the extract compared to control and (*ii*) ratio of JC-1 monomer vs. JC-1 aggregates. (**D**) Total GSH, free GSH, and GSSG levels in extract-treated MCF-7 cells. (**E**) % SOD activity. Data represented as mean ± SD (*n* = 3), **** *p* < 0.0001 vs. control, ns(non-significant) *p* > 0.05.

**Figure 6 molecules-27-08222-f006:**
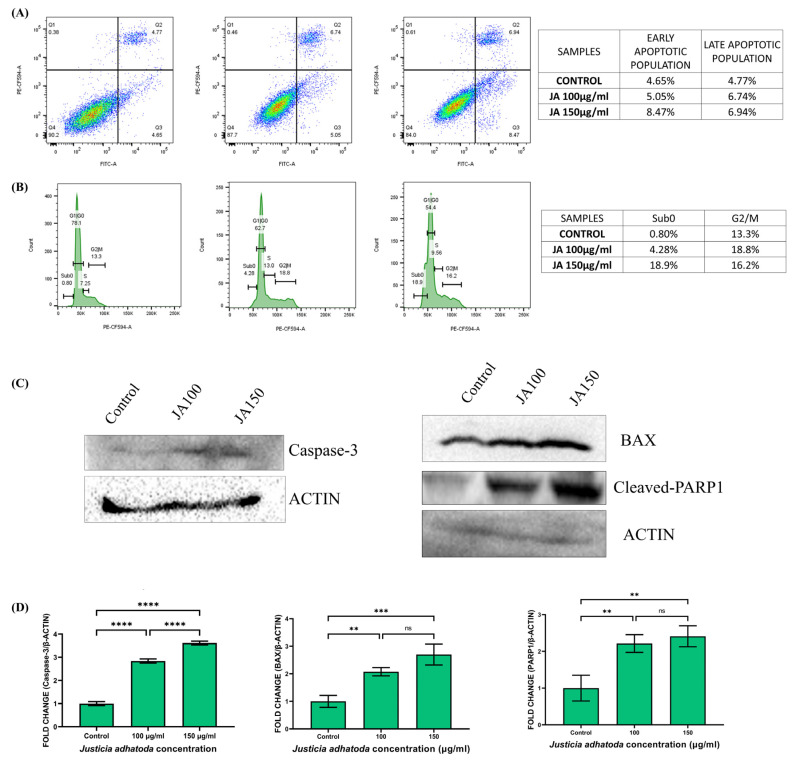
*J. adhatoda* leaf extract induced apoptosis and cell cycle arrest in MCF-7 cells. (**A**) Early and late apoptotic cells in MCF-7 cells treated with 100 μg/mL and 150 μg/mL *J. adhatoda* leaf extract after staining with annexin V/PI. (**B**) The distribution of MCF-7 cells at the various phases of the cell cycle 24 h post-treatment with 100 μg/mL and 150 μg/mL of *J. adhatoda* leaf extract was determined compared to control cells. (**C**) Level of Bax, caspase-3, and cleaved-PARP1 analyzed by Western blotting with specific antibodies; β-actin was used as loading control. (**D**) Representative graph showing expression levels of caspase-3, Bax, and cleaved-PARP1. Data represented as the mean ± SD (*n* = 3) **** *p* < 0.0001 vs. control; *** *p* ≤ 0.001; ** *p* ≤ 0.01 and ns (non-significant), *p* > 0.05.

**Figure 7 molecules-27-08222-f007:**
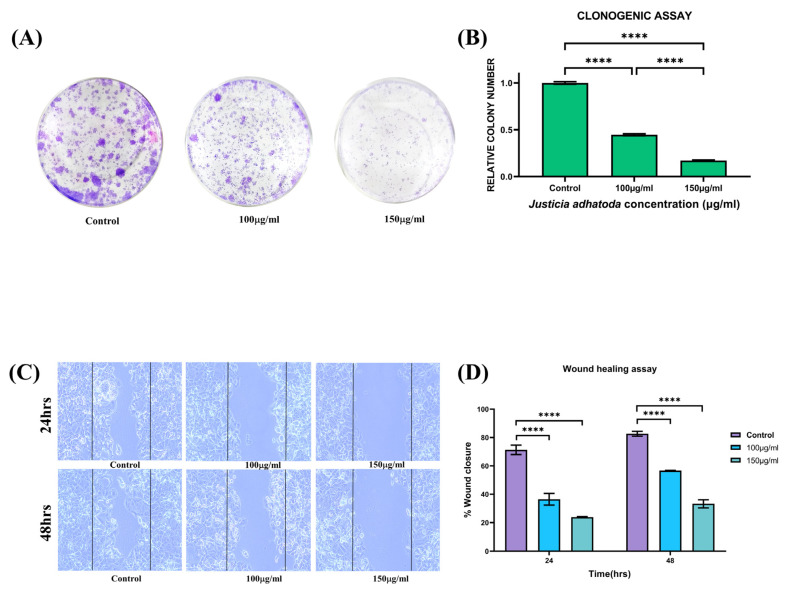
Clonogenic and wound healing assay on MCF-7 cells. (**A**) MCF-7 cells treated with 100 μg/mL and 150 μg/mL of *J. adhatoda* leaf extract showed a significant reduction in the number of colonies as compared to the control group. (**B**) Representative histogram showing the number of colonies. (**C**) Representative images of wound healing after 24 h and 48 h of treatment. (**D**) Representative graph showing % wound closure. Data represented as the mean ± SD of triplicate experiments **** *p* < 0.0001 vs. control.

**Figure 8 molecules-27-08222-f008:**
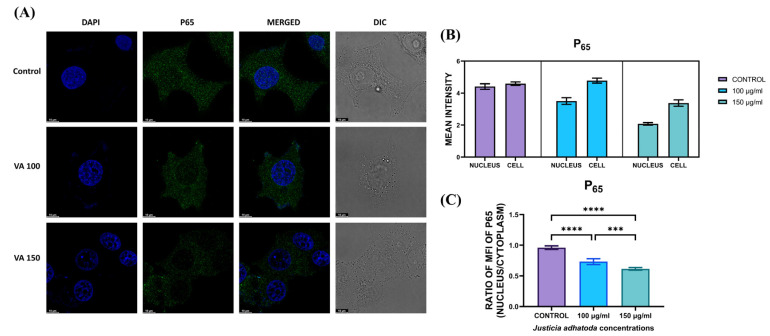
Evaluation of *J. adhatoda* leaf extract on NF-κB p65 translocation. (**A**) Immunofluorescence staining images of p65 translocation in MCF-7 cells post 24 h treatment with 100 μg/mL and 150 μg/mL of *J. adhatoda* leaf extract, measured by confocal microscopy imaging (100· magnification). (**B**) Representative histogram of translocation of p65 into nucleus in MCF-7 cells. (**C**) The histogram representing the ratio of mean fluorescent intensity between nucleus and cytoplasm. Data represented as the mean ± SD. **** *p* < 0.0001 vs. control; *** *p* ≤ 0.001.

**Table 1 molecules-27-08222-t001:** Docking score of the *J. adhatoda* ligands with NF-κB.

Target	PDB ID	Compounds	Binding Energy (k Cal/mol)
**NF-κB**	1LE5	Vasicine	−8.71
		Vasicinol	−8.167
		Vasicinone	−7.604
		Deoxyvasicinone	−7.56
		Vasicinolone	−7.141

## Data Availability

The datasets used and analyzed during the study are available from the corresponding author upon reasonable request.

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
