# Peer review of "In Vitro Anticancer Activity of Methanolic Extract of Justicia adhatoda Leaves with Special Emphasis on Human Breast Cancer Cell Line"

_molecules, 2022, doi:10.3390/molecules27238222_

Round 1

Reviewer 1 Report

Dear Editor and Authors,

Kindly find the attached Comments.

Thank you

Author Response

The authors highly appreciate the insightful comments and suggestions from the learned reviewer. Suggested experiments have been done and the corrections are also made. The authors believe that the revised manuscript have been much improved and would like to thank the reviewers for their valuable comments and suggestions. 

Reviewer 2 Report

The manuscript entitled "In vitro anticancer activity of methanolic extract of Justicia adhatoda leaves with special emphasis on human breast cancer cell line" is interesting and complete. However, for their acceptance in the journal "Molecules" the authors must answer the following questions:

- The authors should determine the antioxidant capacity of the total phenolic and flavonoid content in the extracts by some method FRAP, ABTS, DPPH, ...

- Why was cell viability determined by the MTT method?

- The results obtained in ROS and GSH seem contradictory.

- Do the extracts, at these concentrations, have a pro-oxidant effect?

- What type of apoptosis do extracts from Justicia adhatoda leaves produce? Intrinsic or extrinsic? The authors do not mention it, but modification of the mitochondrial potential is observed.

- How would the increase in ROS affect the translocation of the NF-kB factor to the cell nucleus?

- Clonogenic assay, do they add anything to cell viability studies?

- Cell migration assay should be better explained.

- Taking into account the points made above, the discussion should be rewritten.

Author Response

(The authors gave the same response as above.)

Round 2

Reviewer 1 Report

There are some syntax errors in the manuscript revised manuscript. The authors should check carefully and correct them before publication.

Reviewer 2 Report

The manuscript has been greatly improved and suggestions have been taken into account. However, there are figures that should be improved because their size is very small and they are not well appreciated (Figures 2, 5, 6 and 8). Therefore, once these suggestions are taken into account, the work could be accepted.